# Series/Parallel Switching for Increasing Power Extraction from Thermoelectric Power Generators

**DOI:** 10.3390/mi15081015

**Published:** 2024-08-07

**Authors:** Shingo Terashima, Ryuji Sorimachi, Eiji Iwase

**Affiliations:** 1Department of Applied Mechanics and Aerospace Engineering, Waseda University, 3-4-1 Okubo, Shinjuku, Tokyo 169-8555, Japan; sorimachi@iwaselab.amech.waseda.ac.jp; 2Kagami Memorial Research Institute for Materials Science and Technology, Waseda University, 2-8-26 Nishiwaseda, Shinjuku, Tokyo 169-0051, Japan

**Keywords:** thermoelectric generation, series/parallel switching, energy harvesting, impedance matching

## Abstract

We propose a method for increasing power extraction from a thermoelectric generator (TEG) by switching between series/parallel circuit configurations of thermoelectric elements, which can adjust the internal impedance of the TEG. The power characteristics of the TEG can be adjusted to the load characteristics of the connected device and the relevant ambient temperature. In this paper, we analyzed the change in the TEG characteristics with the series/parallel switching function. We evaluated the power supply to the connected devices at different ambient temperatures and different series/parallel configurations and confirmed that the extracted power could be increased. By theoretically analyzing the circuit configuration of the thermoelectric devices, the switching required to improve the power extraction, and the temperature difference at which switching occurred, we devised a design method for a TEG with circuit switching in order to increase power extraction with any device. We demonstrated the configuration of switching by using a system in which a TEG supplied power to an external wireless transmitter circuit. In this system, the optimal configuration differed at temperature differences of 3.0 K and 4.0 K. At a temperature difference of 3.0 K, the 2-series/1-parallel configuration provided 10% more power to the external circuit than the 1-series/2-parallel configuration. On the other hand, at the temperature difference of 4.0 K, the 1-series/2-parallel configuration provided 23% more power than the 2-series/1-parallel configuration.

## 1. Introduction

The Internet of Things (IoT), in which devices with communication functions are embedded in various machines and structures in the environment and connected to the Internet to exchange information, has attracted great attention. To achieve an IoT society, a huge number of wireless sensors and batteries must be installed. However, it is difficult to manage the batteries that drive wireless sensors. Therefore, extensive research has focused on energy harvesting, which recovers unused energy to generate power for wireless sensors [1,2,3,4]. Thermal energy is a promising energy source for harvesting because it is abundant in the environment as waste heat [5,6,7,8,9,10,11,12,13,14,15,16,17,18,19]. Thermoelectric power generation can generate electricity from thermal energy sources such as the waste heat, and thermoelectric generators (TEGs) are capable of converting the temperature difference that occurs within a thermoelectric (TE) element into electrical energy. Conventional research on TEGs has focused on improving the TEG performance by developing materials with good thermoelectric generation performances [5,6,7] and flexible structures to obtain good thermal contact with curved heat sources [8,9,10,11,12,13,14,15,16,17,18,19]. In addition, since the extracted power depends on the impedance of an external circuit, it is also important to consider the impedance matching of the TEG and the external circuit [19,20]. However, the I-V characteristics of most external circuits are not linear; that is to say, the impedance varies with the supplied voltage. The supplied voltage from a TEG changes with the ambient temperature. Therefore, TEGs with a fixed internal impedance do not always provide the maximum output power in a way that corresponds to various ambient temperatures. Thus, when the TEG is connected to an external circuit, the extracted power is not determined by the performance of the TEG alone.

Several methods have been proposed for increasing the amount of energy supplied from TEGs to external circuits. For instance, research on power management units (PMUs) for energy harvesting from TEGs has considered the use of boost converters and maximum power point tracking (MPPT) [19,20,21,22,23]. If the temperature difference of the TEG frequently and significantly fluctuates at the installation site, a method such as MPPT is preferable. On the other hand, if the temperature difference does not frequently fluctuate, the power consumption of the MPPT is a disadvantage. Therefore, if the temperature difference and load characteristics of the external circuit are known beforehand, a method of “semi-fixing” the internal impedance of the TEG is preferable. For example, the semi-fixing method has advantages in cases where the settings are manually switched between summer and winter because the semi-fixing method does not consume power during power generation. Therefore, in this paper, we propose a semi-fixing method in order to increase the extracted power from the TEG by changing the impedance of the TEG by altering the circuit configuration, i.e., the number of TE elements in series and in parallel. In addition, the TEG with series/parallel switching was fabricated by incorporating a switching circuit into the TEG and the increase in extracted power was demonstrated experimentally by changing the number of modules in parallel and the temperature difference. Finally, using the fabricated TEG, the charging time of the capacitor in the external circuit was measured to show the amount of increase in the power extracted from the TEG.

## 2. Design

Figure 1 shows a conceptual diagram of the TE elements connected in parallel and series configurations and the relationship between the output characteristics of the TEG and the load characteristics of the external circuit. Figure 1a shows the circuit configurations of *m*-series/4-parallel, 2*m*-series/2-parallel, and 4*m*-series/1-parallel as examples. Figure 1b shows the characteristics of the output characteristics and load characteristics of the external circuit for different temperature differences, Δ*T* [K], on TEG and series/parallel configurations. Here, the temperature difference refers to the temperature difference between the hot side and the cold side of the substrate in the TEG. At the same Δ*T*, the open-circuit voltage of the *m*-series/4-parallel configuration (config. A) is lower than that of the 2*m*-series/2-parallel configuration (config. B) and the 4*m*-series/1-parallel configuration (config. C). The intersection points (operating points) with the load characteristics of the external circuit change as the power supply characteristics of the TEG vary with the temperature difference. The intersection point indicates the operating point of the external circuit. The external circuit connected to the TEG is driven using the voltage and power at the operation point. In the case that Δ*T* is large, the extracted power in the *m*-series/4-parallel configuration (config. A) is larger than that in the 2*m*-series/2-parallel configuration (config. B) or 4*m*-series/1-parallel configuration (config. C). On the other hand, in the case that Δ*T* is small, the extracted power in the 2*m*-series/2-parallel configuration (config. B) is higher than that in *m*-series/4-parallel configuration (config. A) or 4*m*-series/1-parallel configuration (config. C). Therefore, the extracted power from the TEG to the external circuit can be increased by selecting the appropriate TE element circuit configuration according to the temperature difference in the TE elements.

As shown in Figure 2, the curve defined by the intersection points of the output characteristic curves of the TEG can be used to divide the region where the extracted power is maximized for each series/parallel configuration in the *P*-*V* diagram of output power *P* and voltage *V* (purple and yellow dashed and dotted lines in Figure 2). We consider the TEG to consist of *n* pairs of TE elements and suppose that a *m_i_*-series/*l_i_*-parallel configuration is possible (*i* = 1, 2, 3, …, *m_i_ × l_i_* = *n*). Here, we define the internal resistance of a single pair of TE elements as *R*_TE_ and the open voltage of a single pair of TE elements as *V*_TE_(Δ*T*), which is a function of the temperature difference, Δ*T*, that occurs inside the TEG. The total internal resistance of the TEG is $\frac{m_{i}}{l_{i}}R_{TE}\left(=\frac{n}{{l_{i}}^{2}}R_{TE}\right)$ and the voltage of the TEG is $m_{i}V_{TE}\left(\Delta T\right)\left(=\frac{n}{l_{i}}V_{TE}\left(\Delta T\right)\right)$. Defining the supplied voltage to the external circuit as *V*, the supplied current to the external circuit is obtained by $\left(\frac{n}{l_{i}}V_{TE}\left(\Delta T\right)-V\right)/\frac{n}{{l_{i}}^{2}}R_{TE}$. Therefore, the output characteristics on the *P*-*V* diagram can be expressed as follows:(1)$$P=\frac{\left(\frac{n}{l_{i}}V_{TE}\left(\Delta T\right)-V\right)V}{\frac{n}{{l_{i}}^{2}}R_{TE}}.$$

Similarly, for a parallel number of *l*_i+1_ and the total number of TE elements of *n*, the output characteristics can be expressed as follows:(2)$$P=\frac{\left(\frac{n}{l_{i+1}}V_{TE}\left(\Delta T\right)-V\right)V}{\frac{n}{{l_{i+1}}^{2}}R_{TE}}.$$

Using Equations (1) and (2), Equation (3) can be obtained by removing the *V*_TE_(Δ*T*) and the boundary line between the regions of parallel numbers. *l*_i_ and *l*_i+1_ can be determined as follows:(3)$$P=\frac{l_{i}\cdot l_{i+1}}{nR_{TE}}V^{2}.$$

Within the regions divided by the boundary line of equation (3), the maximum extracted power can be achieved under the same circuit configuration. Using the *P*-*V* diagram, which is divided by the boundary line determined by Equations (1)–(3), it is possible to determine which series/parallel circuit configuration achieves the maximum extracted power in relation to the load characteristics of the external circuit connected to the TEG. Using the load characteristics of the external circuit in Figure 2b, shown by the black line, the load characteristics pass through only two regions, *i* = 1 and *i* = 2, and cross the boundary line only once. From this result, it can be predicted that the only circuit configuration necessary to obtain the maximum extraction power is a circuit configuration using parallel numbers *l*_1_ and *l*_2_. In addition, regarding the temperature difference, Δ*T*, obtained from the environment, it is possible to establish at which temperature difference the circuit configuration capable of obtaining the maximum power output is switched. Therefore, by using the series/parallel switching function, TEG users can design a model to obtain maximum extracted power.

## 3. Effect on Extracted Power of Switching Circuit Configuration

We designed and fabricated a power supply circuit capable of switching between the 2-series/1-parallel and 1-series/2-parallel circuit configurations. By connecting the two TE modules, we were able to switch between the two circuit configurations without changing the arrangement of the TE elements. The design of the power supply circuit that can switch between the 2-series/1-parallel and 1-series/2-parallel circuit configurations is shown in Figure 3a. By switching between the two TE modules, we could switch between the two circuit configurations without changing the arrangement of the TE elements. The power supply circuit fabricated in this research is shown in Figure 3b. Switching the two switches toward the center gave a 2-series/1-parallel configuration, and switching the two switches toward the edges gave a 1-series/2-parallel configuration. The power supply characteristics were measured for each configuration as well as for various temperature differences in the TE elements. The current of the TE modules was measured with a source meter while sweeping the voltage from 0 to 200 mV at 2.0 mV intervals. We determined the intersection point between the measured power supply characteristics and the load characteristics of the external circuit, and the validity of changing the connection of the TE elements was evaluated. In addition, in order to confirm the increase in the extracted power, the capacitor charge was evaluated. The power supply characteristics of the TEGs were measured under natural convection conditions, and the room temperature was about 293 K during the measurements. The temperature difference in the TE element was changed by adjusting the set temperature of the hot plate, which was a high-temperature heat source. In this experiment, Δ*T* = 3 K and Δ*T* = 4 K were adopted. The temperature differences Δ*T* = 3 K and Δ*T* = 4 K were defined based on the heat source temperature and the outside air temperature in the actual environment. For Δ*T* = 3, the heat source temperature and the outside air temperature were 309 K and 293 K, respectively. For Δ*T* = 4 K, the heat source temperature and the outside air temperature were 319 K and 293 K, respectively. The temperature difference generated by the TEG was measured using two thermocouples. As shown in Figure 3c, one thermocouple was fixed to the interface between the substrate surface and the bottom of the TEG using a high-thermal-conductivity adhesive (CEMEDINE Co., Ltd., Tokyo, Japan, SX-1008), and the other thermocouple was fixed to the interface between the top of the TEG and the heat sink using the high-thermal-conductivity adhesive. Fixing the thermocouple to the interface with a high-thermal-conductivity adhesive overcomes the problems of thermal contact and insulation, enabling highly accurate temperature measurement.

Figure 3d shows the power supply characteristics and the load characteristics of the external wireless transmission circuit when the TEG power supply circuit was switched between the 2-series/1-parallel and 1-series/2-parallel configurations. The external wireless transmitter circuit consists of a boost converter (Coilcraft Inc., Cary, IL, USA, LPR6235-752SMR), a charging capacitor (470 μF) and a wireless transmitter with a microcontroller (Mono Wireless Inc., Kanagawa, Japan, TWELITE RED). From the Figure 3d, we can estimate that the optimal configuration differs for temperature differences of 3.0 K and 4.0 K in this system in which the TEG supplies power to the external wireless transmitter circuit. Table 1 shows the extracted power at each operating point. For a small temperature difference (Δ*T* is 3.0 K) in the TE element, the extracted power was larger by about 18 µW when the 2-series/1-parallel configuration was used than when the 1-series/2-parallel configuration was used. In contrast, for a large temperature difference (Δ*T* is 4.0 K) in the TE element, the extracted power was larger by about 65 µW when the 1-series/2-parallel configuration was used compared to when the 2-series/1-parallel configuration was used. Therefore, the power extracted from the TEGs to the external circuit was increased by switching the series/parallel configuration of the TE elements according to the temperature differences in the TE element.

Finally, the capacitor charge was evaluated to confirm the increase in the extracted power. Figure 3e and Table 2 show the charging amount and charging time from the TEG to the capacitor. The capacitance of the capacitor used in this experiment was 470 μF, and the capacitor was connected to a wireless transmitter. When the capacitor was charged with 3.3 V, the system supplied power to the wireless transmitter. The first charging time, *t*_1_, and the stable charging time, *t*_2_, were faster in the 2-series/1-parallel configuration than in the 1-series/2-parallel configuration when Δ*T* was 3.0 K. In contrast, when Δ*T* was 4.0 K, the charging time for the 1-series/2-parallel configuration was faster than that of the 2-series/1-parallel configuration. Therefore, switching the series/parallel configuration according to the temperature difference increased the extracted power and shortened the charging time from the TEG to the capacitor.

## 4. Conclusions

We designed and fabricated a TEG with a series/parallel switching function to increase the extracted power, and we evaluated the utility of this switching function. First, we theoretically showed which series/parallel configurations maximized the extracted power in which regions in the P-V diagram. Next, we fabricated a TEG with a series/parallel switching function using two TE modules and measured the output characteristics of the TEG and the load characteristics of the external circuit at each operating point. When the temperature difference in the TE element was 3.0 K, the extracted power of the 2-series/1-parallel configuration was larger than that of the 1-series/2-parallel configuration. In contrast, when the temperature difference in the TE element was 4.0 K, the extracted power of the 1-series/2-parallel configuration was higher than that of the 2-series/1-parallel configuration. In addition, we measured the charging time of the capacitor in the external circuit for each series/parallel configuration and showed that the charging time could be shortened by applying an appropriate circuit configuration. Therefore, it is necessary to change the numbers of series and parallel TE elements in order to drive the external circuit, connected to the TEGs, with high efficiency.

## Figures and Tables

**Figure 1 micromachines-15-01015-f001:**
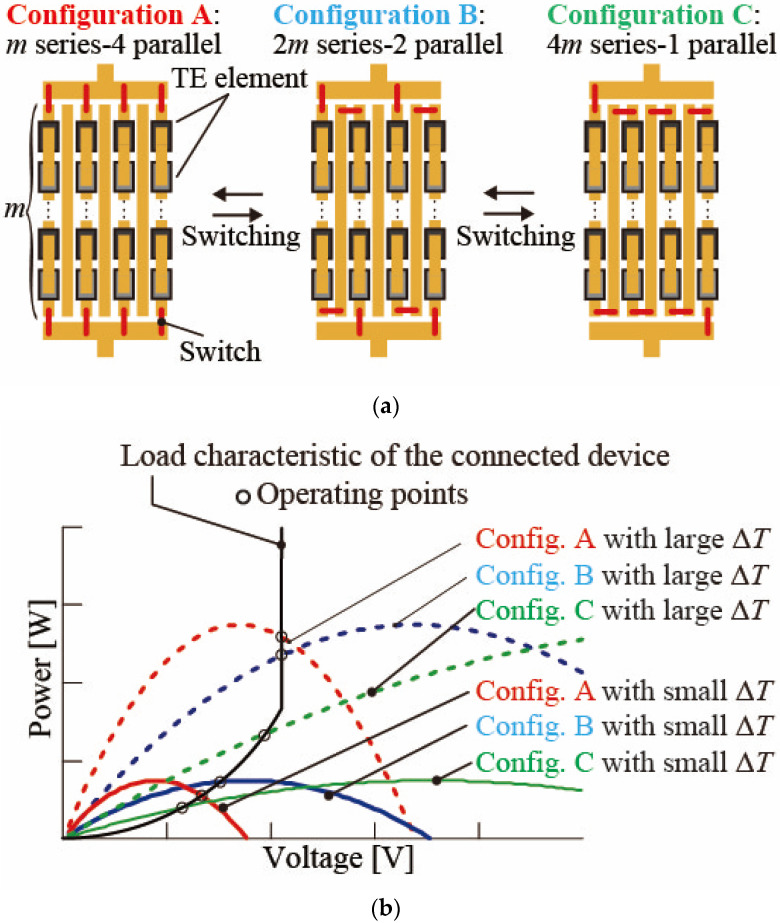
Schematics of the calculated series/parallel configurations and output and load characteristics: (**a**) A schematic of the *m*-series/4-parallel, 2*m*-series/2-parallel, and 4*m*-series/1-parallel configurations of thermoelectric elements. (**b**) The relationship between the load characteristics of the external circuit when the resistance of the output circuit changes depending on the temperature difference of the thermoelectric elements and the power supply characteristics.

**Figure 2 micromachines-15-01015-f002:**
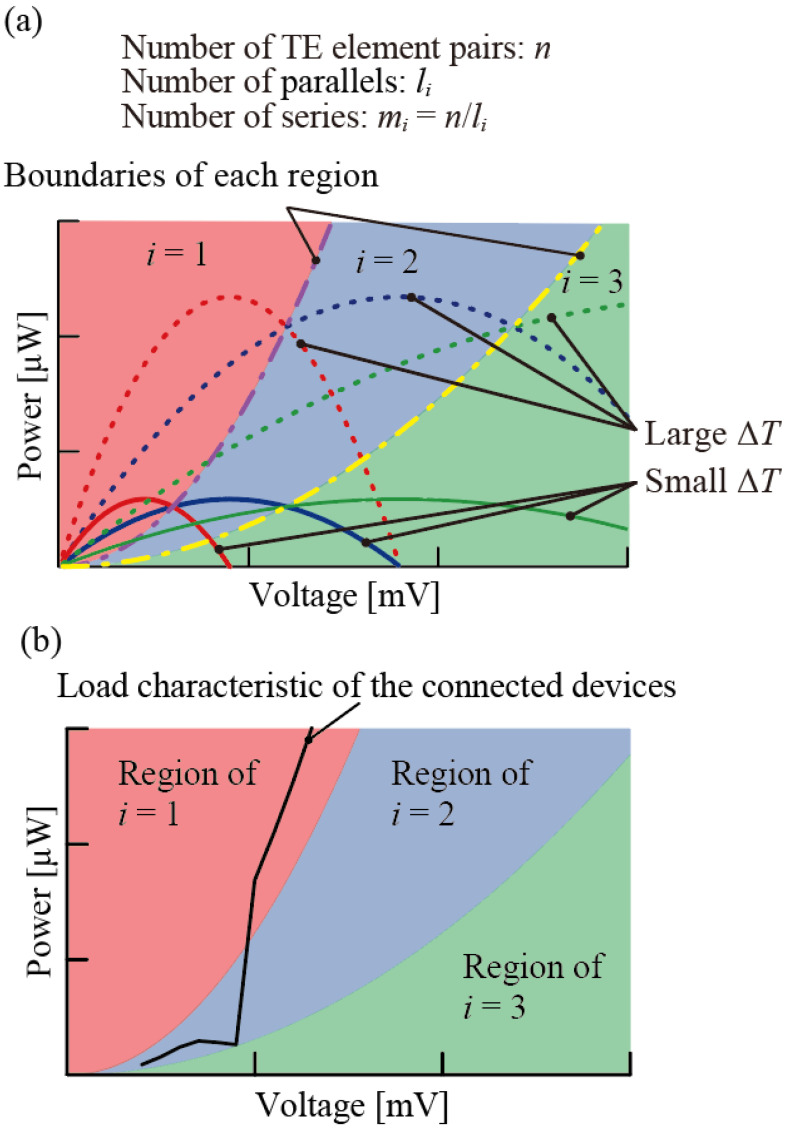
Schematics of area division regarding maximum extracted power: (**a**) Boundary line that separates the regions where certain circuit configuration exhibits maximum extracted power. Red, blue, and green in the figure indicate the parallel numbers *l*_1_, *l*_2_, and *l*_3_, respectively. (**b**) Relationship between load characteristics of connected external circuits and each area.

**Figure 3 micromachines-15-01015-f003:**
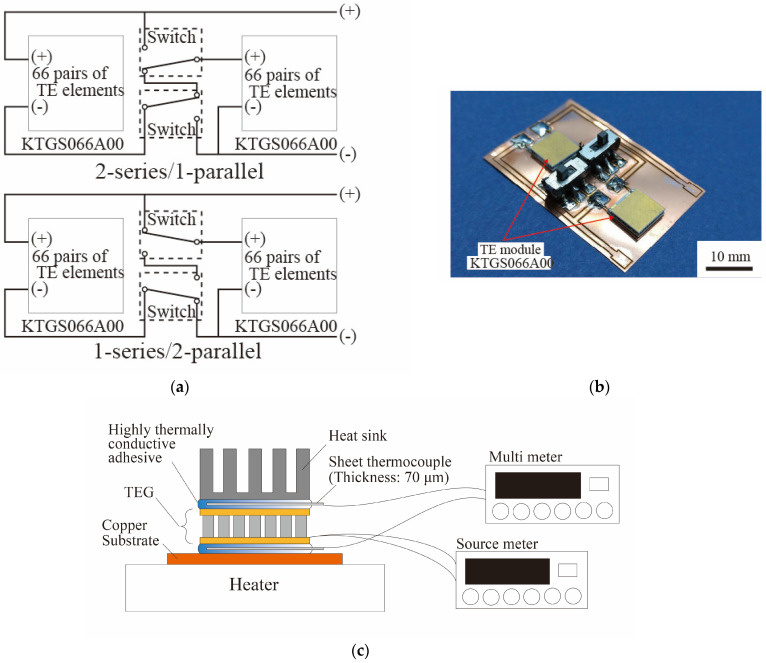
(**a**) Schematics of power supply circuits, (**b**) photograph of fabricated power supply circuit, (**c**) schematics of setup for temperature and power characteristic measurements, (**d**) measured power supply characteristics and load characteristics of external circuit, and (**e**) first charging time and charging time after capacitor becomes stable.

**Table 1 micromachines-15-01015-t001:** Power extracted from the TEG to the external circuit.

	2-Series/1-Parallel	1-Series/2-Parallel
Δ*T* = 3.0 K	OP_S1_: 195 ± 1.3 µW	OP_P1_: 177 ± 1.6 µW
Δ*T* = 4.0 K	OP_S2_: 283 ± 1.5 µW	OP_P2_: 348 ± 2.2 µW

**Table 2 micromachines-15-01015-t002:** Charging time for the capacitor.

	2-Series/1-Parallel	1 Series/2 Parallel
Δ*T* = 3.0 K	*t*_1_: 69.7 s, *t*_2_: 30.6 s	*t*_1_: 78.2 s, *t*_2_: 32.4 s
Δ*T* = 4.0 K	*t*_1_: 66.5 s, *t*_2_: 27.6 s	*t*_1_: 65.7 s, *t*_2_: 27.0 s

*t*_1_ is the first charging time and *t*_2_ is the charging time after the capacitor is stable.

## Data Availability

The data presented in this study are available on request.

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
