# Peer review of "Series/Parallel Switching for Increasing Power Extraction from Thermoelectric Power Generators"

_micromachines, 2024, doi:10.3390/mi15081015_

Round 1

Reviewer 1 Report

Comments and Suggestions for Authors

The work presents a method to boost the power extracted from a TEG module by switching between series and parallel circuit configurations of thermoelectric elements, allowing to modify the internal impedance of the TEG.

I have some suggestions in order to improve the quality of the paper,

1-    Introduction section. The novelties and contributions of the paper need to be further stressed. A possibility is to add a list with these point while exposing the advantages and shortcomings of the proposal with respect the state of the art.

2-    The authors must comment this sentence found in page 1: “TEGs with fixed internal impedance cannot always achieve the maximum output power corresponding to various ambient temperatures”. Using maximum power point tracking can solve this problem, can't it?

3-    Figure 1. The comments and curves in this figure correspond to the TEG and the load with no MPPT. The authors are suggested to comment about the change whwn introducing a MPPT device.

4-    Line 80: “and the output power of the TE is” should read voltage instead of power.

5-    Table 2 gives the charging time of the capacitor and t1 and t2.It is not clear which capacitor is it. Please add a circuit diagram with the capacitor and its value and then clearly specify t1 and t2 (already done in Figure 3).

6-    Line 125. “The temperature difference generated in the TEG was measured using two thermocouples: one attached to the bottom surface of the TEG and the other attached to the top surface.” Thermocouples are not very accurate and you are measuring small temperature differences. It is necessary to add the full characteristics of the thermocouples used including their inaccuracies in the measured temperature range.

7-    An efficiency analysis under the different analyzed operating conditions would be valuable.

8-    Please add more state of the art references from 2023-2024.

I hope the remarks above can help to improve the quality and readability of this research work.

Author Response

Comments 1:[Introduction section. The novelties and contributions of the paper need to be further stressed. A possibility is to add a list with these point while exposing the advantages and shortcomings of the proposal with respect the state of the art.]

Response 1:[Thank you for your advice. The novelty of this paper is the ability to improve the driving efficiency of external circuits connected to thermoelectric generator by simply changing the number of series and parallel connections, without enhancing the performance of the thermoelectric materials or modifying the device structure. However, for thermoelectric generator installed in locations where the heat source temperature and ambient temperature fluctuate significantly, the advantages of this series-parallel switching design are not realized. This has been added to the paper from lines 64 to 77.]

Comments 2:[The authors must comment this sentence found in page 1: “TEGs with fixed internal impedance cannot always achieve the maximum output power corresponding to various ambient temperatures”. Using maximum power point tracking can solve this problem, can't it?]

Response 2:[Thank you for pointing this out. As you said, MPPT (Maximum Power Point Tracking) is a method for efficiently supplying power to external circuits. However, it has two problems: 1) initial cost and 2) power consumption. Regarding 1) initial cost, MPPT controllers are a significant cost burden for thermoelectric power generation devices, which are inexpensive compared to solar and wind power generators, so the series-parallel switching design proposed in this paper is superior to MPPT. Regarding 2) power consumption, the MPPT controller needs to constantly perform calculations, so it constantly consumes the power generated by the thermoelectric power generation device. On the other hand, the series-parallel switching design is superior to MPPT because it does not consume power. I added this to the introduction of my paper.]

Comments 3:[Figure 1. The comments and curves in this figure correspond to the TEG and the load with no MPPT. The authors are suggested to comment about the change with introducing a MPPT device.]

Response 3:[As you said, Figure 1 does not show the changes before and after the efficiency improvement by MPPT. Figure 1 explains the series/parallel switching method proposed in this research.]

Comments 4:[Line 80: “and the output power of the TE is” should read voltage instead of power.]

Response 4:[Thank you for pointing that out. As you said, I have corrected it to voltage instead of output power.]

Comments 5:[Table 2 gives the charging time of the capacitor and t1 and t2. It is not clear which capacitor is it. Please add a circuit diagram with the capacitor and its value and then clearly specify t1 and t2 (already done in Figure 3).]

Response 5:[Thank you for pointing that out. I have added the following text to the paper. "The external wireless transmitter circuit is consist of a boost converter (LPR6235-752SMR, Coilcraft), a charging capacitor (470 μF), and a wireless transmissior with microcontroller (TWELITE RED, Mono Wireless). From the Figure 3(c), we can estimate that the optimal configuration differs for temperature differences of 3.0 K and 4.0 K in this system in which the in which the TEG supplies power to the external wireless transmitter circuit."]

Comments 6:[Line 125. “The temperature difference generated in the TEG was measured using two thermocouples: one attached to the bottom surface of the TEG and the other attached to the top surface.” Thermocouples are not very accurate and you are measuring small temperature differences. It is necessary to add the full characteristics of the thermocouples used including their inaccuracies in the measured temperature range.]

Response 6:[Thank you for your comments. When measuring temperature using the thermocouple, there are concerns about (1) thermal contact with the heat source surface and (2) insulation. Concerning (1), the thermocouple used is very thin (70 μm thick), which ensures excellent thermal contact, resulting in temperature fluctuations of only ±0.02°C. Regarding (2), the thermocouple is positioned between the high-temperature side of the thermoelectric device and the high-temperature heat source (or between the low-temperature side of the thermoelectric device and the heat sink), providing high insulation from the external environment. This information has been added to Chapter 3.]

Comments 7:[An efficiency analysis under the different analyzed operating conditions would be valuable.]

Response 7:[Thank you for pointing this out. As you suggested, I believe experiments under other conditions would be valuable. However, this paper primarily focuses on the series-parallel switching design. Therefore, demonstrating the effectiveness of the series-parallel switching design proposed in this paper is crucial. The paper shows changes in operating points at temperature differences (ΔT=3 K and 4 K) where the optimal series-parallel switch occurs, along with the supply time to the capacitor. The temperature differences ΔT=3 K and 4 K chosen in this paper precisely mark the switch points. I added this to my paper.]

Comments 8:[Please add more state of the art references from 2023-2024.]

Response 8:[Thank you. As you suggested, I have added references discussing methods to improve the efficiency of thermoelectric generator published from 2023 to 2024.]

Reviewer 2 Report

Comments and Suggestions for Authors

The paper proposes a method to increase power extraction from a thermoelectric generator (TEG) by switching between series/parallel circuit configurations of thermoelectric elements. This switching adjusts the TEG's internal impedance to better match the load characteristics and ambient temperature variations. Experimental results demonstrate that by selecting the appropriate configuration based on temperature differences, the power output and efficiency of the TEG can be significantly improved. Overall, the paper makes a significant contribution to the field of thermoelectric energy harvesting. However, I think the manuscript should be improved. My suggestions are below:

  • Line 14-15: “The power extracted from the TEG is increased because the power characteristics of the TEG can be adjusted to the load characteristics of the connected device and the ambient temperature.” This sentence should be checked. Such expressions are unnecessary in the summary and there is a semantic error in the sentence.
  • Give some important numerical results in “Abstract” section.
  • Line 46: It is said “internal impedance”…I think writing only “impedance” is sufficient to define internal resistance.
  • Explain more detailed the meaning of Figure 1(b). What is the value of small and large temperature differences? What do you mean with illustrating the operating points?
  • The formulas of (1) and (2) should be explained with related references.
  • Do you mean 3K and 4K as small and large temperature differences which was given in tables? If yes, how appropriate is it to use definitions such as small and large for such a small temperature difference?
  • In this context, as said before the experiments cover a narrow range of temperature differences (3.0 K and 4.0 K). While these conditions demonstrate the feasibility of the approach, they may not fully represent the wide range of operating conditions in practical applications. Expanding the temperature range would provide a more comprehensive evaluation.
  • The data analysis could be more comprehensive. For instance, incorporating statistical significance tests of the observed differences in power extraction between configurations would provide a more rigorous validation of the results.
  • Additionally, analyzing the long-term stability and performance of the switching mechanism would be beneficial.
  • The novelty of this study should be emphasized by discussing the advantages and potential limitations of the proposed method relative to other impedance matching techniques.

·        The references are relevant, but the literature review could be strengthened by including more recent studies. Specifically, citing recent work on thermoelectric materials and dynamic impedance matching would provide a broader context and highlight the relevance of the proposed method. For example, evaluations by referring to the " Ozbektas, S., Kaleli, A., & Sungur, B. (2024). Prediction of the effect of load resistance and heat input on the performance of thermoelectric generator using numerical and artificial neural network models. Applied Thermal Engineering, 249, 123417." study in the current literature, which is closely related to this study, will be useful.

·        Line 154-155: It is said that “When the temperature difference in the TE element was 3 K, the extracted power of the 2- series/1-parallel configuration was larger than that of the 1-series/2-parallel configuration.”. However I see opposite of this information from Figure 3(c).

·        The work is good, but it should be expanded a little more. Increasing analyzes or experiments would be valuable.

  • The study focuses on a small-scale prototype with only two TE modules. The scalability of this method to larger systems with more complex configurations is not addressed. What should authors say the performance and feasibility of the proposed method in larger, real-world systems?

·         

Author Response

Comments 1:[Line 14-15: “The power extracted from the TEG is increased because the power characteristics of the TEG can be adjusted to the load characteristics of the connected device and the ambient temperature.” This sentence should be checked. Such expressions are unnecessary in the summary and there is a semantic error in the sentence.]

Response 1:[Thank you for pointing this out. I have revised the part you pointed out.]

Comments 2:[Give some important numerical results in “Abstract” section.]

Response 2:[Thank you. I also think that numerical statements are important. I have added a sentence containing numbers to the abstract (regarding the increase in power supply).]

Comments 3:[Line 46: It is said “internal impedance”…I think writing only “impedance” is sufficient to define internal resistance.]

Response 3:[Thank you for pointing this out. As you mentioned, I have corrected 'internal impedance' to 'impedance'.]

Comments 4:[Explain more detailed the meaning of Figure 1(b). What is the value of small and large temperature differences? What do you mean with illustrating the operating points?]

Response 4:[Thank you for pointing this out. The temperature difference here refers to the difference in temperature between the high-temperature side and the low-temperature side of the thermoelectric generator (TEG). Specific temperature values are not included in Fig. 1, as it is a schematic diagram. Regarding the operation point, the external circuit connected to the TEG is driven using the voltage and power at the operation point. We have added an explanation of the specific switching methods in the latter half of the introduction.]

Comments 5:[The formulas of (1) and (2) should be explained with related references.]

Response 5:[Thank you for your comment. To make it easier for readers to understand, we have revised the relevant sections to include explanations from Equation (1) to the final equation, along with additional explanations regarding Equation (1).]

Comments 6:[Do you mean 3K and 4K as small and large temperature differences which was given in tables? If yes, how appropriate is it to use definitions such as small and large for such a small temperature difference?]

Response 6:[Thank you for your comment. As you correctly pointed out, ΔT=3 K corresponds to a small temperature difference, and ΔT=4 K corresponds to a larger temperature difference. This has been added to the relevant section.]

Comments 7:[In this context, as said before the experiments cover a narrow range of temperature differences (3.0 K and 4.0 K). While these conditions demonstrate the feasibility of the approach, they may not fully represent the wide range of operating conditions in practical applications. Expanding the temperature range would provide a more comprehensive evaluation.]

Response 7:[Thank you for pointing this out. As you suggested, I believe experiments under other conditions would be valuable. However, this paper primarily focuses on the series-parallel switching design. Therefore, demonstrating the effectiveness of the series-parallel switching design proposed in this paper is crucial. The paper shows changes in operating points at temperature differences (ΔT=3 K and 4 K) where the optimal series-parallel switch occurs, along with the supply time to the capacitor. The temperature differences ΔT=3 K and 4 K chosen in this paper precisely mark the switch points. We have added a note about this in Chapter 3.]

Comments 8:[The data analysis could be more comprehensive. For instance, incorporating statistical significance tests of the observed differences in power extraction between configurations would provide a more rigorous validation of the results.]

Response 8:[Thank you for pointing this out. The error for each output was added to Table 1.]

Comments 9:[Additionally, analyzing the long-term stability and performance of the switching mechanism would be beneficial.]

Response 9:[Thank you for your comment. The series-parallel switching design proposed in this paper does not assume that the external environment (heat source temperature and outside air temperature) will change during the power generation, so we believe that stability evaluation through long-term testing is not necessary. We have added a note in the introduction that the series-parallel switching design proposed in this paper is applicable to places where the heat source temperature and the outside air temperature hardly change.]

Comments 10:[The novelty of this study should be emphasized by discussing the advantages and potential limitations of the proposed method relative to other impedance matching techniques.]

Response 10:[Thank you for pointing this out. This has been added from line 54 to 65 of the introduction.]

Comments 11:[Line 154-155: It is said that “When the temperature difference in the TE element was 3 K, the extracted power of the 2- series/1-parallel configuration was larger than that of the 1-series/2-parallel configuration.”. However I see opposite of this information from Figure 3(c).]

Response 11:[Thank you for pointing that out. However, I think it is consistent with the current text and Figure 3(c). We should have emphasized in the text that "please look at the intersection point with the load characteristics of the external circuit, not the maximum value of each power generation characteristic curve." We apologize for that.]

Comments 12:[The study focuses on a small-scale prototype with only two TE modules. The scalability of this method to larger systems with more complex configurations is not addressed. What should authors say the performance and feasibility of the proposed method in larger, real-world systems?]

Response 12:[Thank you for your comment. Even for larger systems with many TE modules connected, the series-parallel switching design proposed in this paper assumes that the heat source temperature and ambient temperature remain nearly constant. Therefore, we believe that the results of this paper will not change. Additionally, it is not necessary to fabricate a large-scale device using many TE modules. The reason is that the two TE modules used in this experiment can generate sufficient power for wireless transmission.]

Reviewer 3 Report

Comments and Suggestions for Authors

The authors propose a method to increase power extraction from a thermoelectric generator (TEG) by switching between series/parallel circuit configurations of thermoelectric elements, which can adjust the internal impedance of the TEG.

Questions:

1. The literature review is not good. What are the recurring and current works in the literature that have dealt with this problem? What are the weaknesses of existing methods and what do the authors aim to improve or solve?

2. As the bibliographical review is not well done and the introduction is very succinct, it is not clear what the authors' contribution is to what already exists in the literature?

3. It is not clear mathematically how different types of switching provide different levels of power to the load. Presenting only Figure 1 is not enough. Furthermore, the authors do not even present a reference to support this reasoning.

4. If different types of switching can provide higher levels of power, why is it not used? Is there a standard configuration that is used in the literature?

5. A serious problem with the article is that it does not present the reader with a step-by-step algorithm or flowchart of the method proposed by the authors. There is a paragraph discussing it, but it is confusing. How are the switches carried out to achieve the greatest power? What criteria are used?

6. The authors do not present comparative results with methods already existing in the literature. Comparative results allow evaluating the benefits and weaknesses of existing methods.

7. There are few case studies and few discussions. Is it difficult to evaluate the benefits of the authors' contribution in light of everything that already exists in literature?

8. The fact that there is no good literature review, the proposed method is not presented, there are no comparative results with methods already existing in the literature and few case studies and discussions make the article's conclusions a little inconsistent. The article needs to be improved.

Author Response

Thank you for your review. Your valuable comments have been incredibly helpful in improving the quality of my paper.

Comments 1:[The literature review is not good. What are the recurring and current works in the literature that have dealt with this problem? What are the weaknesses of existing methods and what do the authors aim to improve or solve?]

Response 1:[Thank you for pointing this out. MPPT (Maximum Power Point Tracking) is a method for efficiently supplying power to external circuits. However, it has two problems: 1) initial cost and 2) power consumption. Regarding 1) initial cost, MPPT controllers are a significant cost burden for thermoelectric power generation devices, which are inexpensive compared to solar and wind power generators, so the series-parallel switching design proposed in this paper is superior to MPPT. Regarding 2) power consumption, the MPPT controller needs to constantly perform calculations, so it constantly consumes the power generated by the thermoelectric power generation device. On the other hand, the series-parallel switching design is superior to MPPT because it does not consume power. I added this to the introduction of my paper.]

Comments 2:[As the bibliographical review is not well done and the introduction is very succinct, it is not clear what the authors' contribution is to what already exists in the literature?]

Response 2:[Thank you for your comment. There is a method called MPPT (Maximum Power Point Tracking) to increase the efficiency of power supplied to external circuits. MPPT is used mostly when there are significant fluctuations in the heat source temperature or ambient temperature. On the other hand, the series-parallel switching design proposed in this paper is used when the heat source temperature and ambient temperature remain relatively constant. Additionally, the series-parallel switching design does not consume power during TEG generation and is very inexpensive, which promotes early adoption of IoT. This point has been added to the introduction in my paper.]

Comments 3:[It is not clear mathematically how different types of switching provide different levels of power to the load. Presenting only Figure 1 is not enough. Furthermore, the authors do not even present a reference to support this reasoning.]

Response 3:[Thank you for your comments. I have added several recently published papers on improving the efficiency of power supply to external circuits to the references. Additionally, I have included a comparison between existing methods and the proposed series-parallel switching design in the introduction of my paper.]

Comments 4:[If different types of switching can provide higher levels of power, why is it not used? Is there a standard configuration that is used in the literature?]

Response 4:[Thank you for your comment. There has been a lot of research on thermoelectric materials and flexible structures, but until now, there has been no research on improving efficiency through circuit switching. As mentioned above, while MPPT exists, it is not suitable for small and low-cost TEG. Additionally, commercially available TEGs do not have switching circuits and all thermoelectric elements are connected in series. This is the current standard for TEGs.]

Comments 5:[A serious problem with the article is that it does not present the reader with a step-by-step algorithm or flowchart of the method proposed by the authors. There is a paragraph discussing it, but it is confusing. How are the switches carried out to achieve the greatest power? What criteria are used?]

Response 5:[Thank you for your comment. The process is simple: before installation, you need to understand the temperature difference and the load characteristics of the external circuit (such as sensors). Then, based on that temperature difference, manually switch the configuration to maximize the power supply to the external circuit, and install the device. No calculations are needed after installation.]

Comments 6:[The authors do not present comparative results with methods already existing in the literature. Comparative results allow evaluating the benefits and weaknesses of existing methods.]

Response 6:[Thank you for pointing this out. Compared to the existing method of MPPT (Maximum Power Point Tracking), there are two main differences in terms of initial cost and power consumption. MPPT controllers are expensive, so using them with solar or wind generators, which are already costly, doesn't significantly change the initial cost while improving efficiency. However, for relatively inexpensive TEGs, MPPT controllers represent a significant cost burden, making the series-parallel switching design proposed in this paper superior to MPPT. MPPT controllers require calculations during power generation, continuously consuming power generated by the TEG. On the other hand, the series-parallel switching design does not consume power, making it more advantageous than MPPT.]

Comments 7:[There are few case studies and few discussions. Is it difficult to evaluate the benefits of the authors' contribution in light of everything that already exists in literature?]

Response 7:[Thank you for your comment. MPPT is advantageous when the source temperature and ambient temperature frequently fluctuate. On the other hand, the series-parallel switching design proposed in this paper is advantageous when the source temperature and ambient temperature do not change frequently. In other words, it benefits "semi-fixing" applications where settings are adjusted seasonally, such as between summer and winter.]

Comments 8:[The fact that there is no good literature review, the proposed method is not presented, there are no comparative results with methods already existing in the literature and few case studies and discussions make the article's conclusions a little inconsistent. The article needs to be improved.]

Response 8:[Thank you for pointing this out. We have added a comparison with existing efficiency improvement methods and discussed the strengths and contributions of the switching design proposed in this paper. Additionally, we have included references to the literature on existing methods in my paper.]

Round 2

Reviewer 1 Report

Comments and Suggestions for Authors

The authors have replied my questions

Comments on the Quality of English Language

There are some errors. The authors are suggested to proofread the manuscript.

Author Response

Comments 1:[The authors have replied my questions. There are some errors. The authors are suggested to proofread the manuscript.]

Response 1:[I appreciate your evaluation of my responses to the reviewers' questions. Thank you very much.I'll proofread my English thoroughly with a native speaker. In particular, several meaningless spaces were found in the English words, which were corrected. As a result of English proofreading, the corrected parts have been marked in yellow.]

Reviewer 2 Report

Comments and Suggestions for Authors

As said before the experiments cover a narrow range of temperature differences (3.0 K and 4.0 K). While these conditions demonstrate the feasibility of the approach, they may not fully represent the wide range of operating conditions in practical applications. The authors stated that they focused on series and parallel connections. However giving these temperature differences must be based on a a more meaningful physical basis. There are no big differences in terms of TEG between 3K and 4K. Also, what is the accuracy of thermocouples? The connection of the thermocouples also should be illustrated? 

Author Response

Comments 1:[As said before the experiments cover a narrow range of temperature differences (3.0 K and 4.0 K). While these conditions demonstrate the feasibility of the approach, they may not fully represent the wide range of operating conditions in practical applications. The authors stated that they focused on series and parallel connections. However giving these temperature differences must be based on a a more meaningful physical basis. There are no big differences in terms of TEG between 3K and 4K. Also, what is the accuracy of thermocouples? The connection of the thermocouples also should be illustrated? ]

Response 1:[Thank you for pointing this out. I'm sorry, I didn't mention the physical basis for the temperature. The temperature differences ΔT=3 and ΔT=4 K described in this paper were defined based on the heat source temperature and outside air temperature in a real environment. With ΔT=3, the heat source temperature and the outside air temperature were 36°C and 20°C, respectively. With ΔT=4K, the heat source temperature and the outside air temperature were 46°C and 20°C, respectively. Even if the temperature difference between the heat source temperature and the outside air temperature is large, a large ΔT is not achieved because the thermoelectric power generation module is thin. The temperatures 36°C and 46°C adopted here correspond to the temperatures of most heat sources present in the environment. For example, 36°C corresponds to human body temperature. This was added to the paper at lines 163 to 168. Regarding the measurement accuracy of thermocouples, when measuring temperature using the thermocouple, there are concerns about (1) thermal contact with the heat source surface and (2) insulation. Concerning (1), the thermocouple used is very thin (70 μm thick), which ensures excellent thermal contact, resulting in temperature fluctuations of only ±0.02°C. Regarding (2), the thermocouple is positioned between the high-temperature side of the thermoelectric device and the high-temperature heat source (or between the low-temperature side of the thermoelectric device and the heat sink), providing high insulation from the external environment. This information has been added to the paper at lines 169 to 175. In addition, to make it easier for readers to understand, we have added new figure of thermocouple connections in the paper.]

Reviewer 3 Report

Comments and Suggestions for Authors

The authors propose a method to increase power extraction from a thermoelectric generator (TEG) by switching between series/parallel circuit configurations of thermoelectric elements, which can adjust the internal impedance of the TEG.

The article has been improved, the contribution is good and all questions have been effectively answered.

Author Response

Comment 1:[The authors propose a method to increase power extraction from a thermoelectric generator (TEG) by switching between series/parallel circuit configurations of thermoelectric elements, which can adjust the internal impedance of the TEG. The article has been improved, the contribution is good and all questions have been effectively answered.]

Response 1:[I appreciate your positive evaluation of my responses to the reviewers' questions. Thank you very much.]

Round 3

Reviewer 2 Report

Comments and Suggestions for Authors

All necessary corrections were made by the authors.